# Does environmental information disclosure promote enterprise green technology innovation?

Kun Zhang[1], Ji Li[2], Weigang Ma[2], Xingqi Wang[2]*

1 School of Economics, Zhongnan University of Economics and Law, Wuhan, China, 2 School of Economics and Management, Shihezi University, Shihezi, China

* xingqiwang1030@163.com

**Data Availability Statement:** The relevant data of this study has been published in openICPSR, project number: openicpsr-206002. https://www.openicpsr.org/openicpsr/project/206002/version/V1/view.

## Abstract

Accelerating green technology innovation is essential for promoting economic transformation and achieving sustainable development. Environmental information disclosure (EID) at the city level, as a crucial component of the environmental governance system, provides new opportunities to drive enterprise green technology innovation. This study utilizes the Pollution Source Supervision Information Disclosure Index (PITI), implemented in China since 2008, as a quasi-natural experiment for EID. By integrating data from Chinese A-share listed companies spanning the period from 2003 to 2020, a multi-period difference-in-differences (DID) model is employed to identify the influence of EID. The findings indicate a positive association between EID and enterprise green technology innovation, resulting in concurrent enhancements in both quantity and quality. The robustness of the conclusions remains intact even after addressing endogeneity concerns. Mechanism analysis reveals that EID stimulates environmental governance by facilitating public participation-based, command-control, and market-incentive environmental regulation, thereby fostering enterprise green technology innovation. In addition, the impact of EID on enterprise green technology innovation is heterogeneous, with the policy effect being more pronounced in highly marketized and resource-rich samples. Finally, combining theoretical analyses and empirical results, relevant suggestions are made for formulating more flexible environmental regulatory policies and building a diversified environmental governance system.

## 1. Introduction

China has attained noteworthy economic accomplishments during the past four decades of reform and opening up. However, the relentless pursuit of economic growth, at the expense of other considerations, resulted in a neglect of the quality of economic development [1, 2]. On the one hand, environmental pollution and resource scarcity emerge as formidable challenges confronting China, hampering economic progress. On the other hand, long-standing path dependency engenders a "technological lock-in," where the impetus for technology innovation in economic advancement remains woefully inadequate, impeding sustainable advancement.

**Funding:** This research was funded by the National Social Science Fund of China (23BTJ003). The funding was obtained by Weigang Ma.

**Competing interests:** The authors have declared that no competing interests exist.

In this critical juncture, as countries gravitate towards environmentally friendly and low-carbon development, striking a balance between the environment and development has become an inescapable issue that demands urgent attention from China and other global manufacturing powerhouses. By drawing insights from the experiences of developed nations, we can discern that environmental regulation serves as an effective governance tool in harmonizing the dual objectives of environmental preservation and economic growth. At the same time, green technology innovation represents a momentous breakthrough in transcending the traditional trade-off between these two aspects [3, 4]. Therefore, formulating appropriate environmental regulatory policies to drive the development of green technological innovation in an economy requires urgent attention from China and other global manufacturing powerhouses.

Green technology innovation is well known to be pivotal in sustainable development [5]. Green technological innovation can improve energy efficiency and reduce pollution emissions [6, 7]. Moreover, green technological innovation enhances environmental performance while enabling firms to achieve sustainable economic performance [8]. The positive impacts of increased innovation capacity in improving the technological sophistication and total factor productivity of enterprises' exports have enabled them to enhance their market competitiveness and expand their export scale [4, 9, 10]. In addition, this innovation can also drive the upgrading of industrial structure, overcome the resource curse, accelerate the optimization of regional energy structure, and drive the transformation of economic structure [11–13]. A consensus exists regarding the urgency and importance of accelerating green technology innovation [14]. However, the externality of innovation activities and the uncertainty of inputs and outputs have hindered firms' willingness to innovate [2, 15]. Moreover, green technology innovation is a gradual, long-term process that requires consistent capital injections [14]. When facing financial constraints or external uncertainty shocks, green technology innovation initiatives, firms will tend to squeeze out R&D investment [16]. Zhong (2022) emphasizes that the solution to problems such as externalities in green technological innovation must rely on an equitable intervention mechanism capable of securing compensation for innovation through government intervention [3]. It is a common practice globally to use environmental regulation and other means to guide or mandatorily regulate corporate innovation.

Scholars generally acknowledge the pivotal role of environmental regulation in pollution mitigation and sustainable development [3, 17, 18]. Nevertheless, divergent perspectives exist regarding the impact of environmental regulation on innovation. On the one hand, proponents of the "Porter Hypothesis" argue that environmental regulation stimulates innovation [16], while on the other hand, views stemming from neoclassical economics contend that environmental regulation hampers innovation [19]. As environmental regulatory mechanisms evolve and advance, environmental information disclosure (EID) has progressively become an effective supplement to formal environmental regulations such as command-control and market-incentive [20, 21]. Therefore, it is worthwhile to explore whether the practice of EID can promote enterprise green technology innovation, which posits it as an "appropriate" form of environmental regulation in line with the propositions of the "Porter Hypothesis." In China, the legalization of environmental governance commenced relatively late, particularly concerning the practice of EID. While the United States had already established the Toxic Release Inventory (TRI) based on the *Emergency Planning and Community Right-to-Know Act* (EPCRA) as early as 1986 [22], China lagged by more than 20 years. Only on May 1, 2008, China implemented the *Measures for Measures on Open Environmental Information (Trial)*, signaling the initiation of EID practices within the country. The objective is to establish a diverse environmental governance system that fosters collaboration among the government, enterprises, organizations, and the public by disclosing environmental pollution and governance information. However, during the initial stages of implementation, there were

challenges, such as the limited scope of disclosure subjects and ambiguity in distinguishing between disclosure and non-disclosure. To assess the efficacy of EID by local governments and safeguard the information needs and rights of various stakeholders, the Institute of Public and Environmental Affairs (IPE) in China, in partnership with the Natural Resources Defense Council (NRDC) in the United States, jointly introduced the Pollution Information Transparency Index (PITI) in 2008. Initially encompassing 113 Chinese cities, the index expanded to 120 cities in 2013, representing the first comprehensive evaluation of EID in Chinese urban areas. It garnered considerable attention and significantly impacted society [23]. The PITI effectively curbed arbitrary exercise of discretionary power by local governments in EID, promoting a more scientific and reasonably informed EID [24, 25]. Additionally, it provided a quasi-natural experiment for scholars to assess the policy effects of EID methodically.

Studies have substantiated the efficacy of EID in pollution control and environmental performance enhancement [25, 26]. However, further advancements are necessary in researching the impact of EID on green technology innovation. Many scholars have verified the positive influence of EID on green technology innovation at the city level [1, 16]. Researchers have also explored the role of enterprises' EID in driving green technology innovation [27]. However, there still needs to be more literature investigating the specific mechanisms through which EID affects enterprise green technology innovation, despite some studies approaching it from social responsibility [28], political pressure, and law enforcement perspectives [29]. In summary, while the existing research on this topic is expanding, areas still warrant improvement. Firstly, most studies tend to simplify the original multi-period difference-in-differences (DID) design to a single-period DID when evaluating the overall impact of EID. This oversimplification compromises the rigor of the research design. Secondly, only some studies have explored establishing a diversified environmental governance system through EID. These potential limitations underscore the significance of this study. Specifically, based on data from Chinese listed companies from 2003 to 2020, this study examines the implementation effects of EID from the perspective of enterprise green technology innovation. The release of the PITI is treated as an exogenous shock of EID. The findings demonstrate a positive relationship between EID and the quantity and quality of enterprise green technology innovation. Mechanistically, EID reinforces public participation-based and government-led environmental regulations, thereby stimulating enterprises' drive toward green technology innovation. Lastly, the effects of EID on enterprise green technology innovation exhibit heterogeneity, with highly marketized and resource-endowed samples reaping more excellent benefits.

This study offers several key contributions. First, a multi-period DID model is employed to assess the overall impact of EID on enterprise green technology innovation. Previous studies have explored the relationship between EID and green innovation, often using single-period DID models [28]. This study, however, accounts for the effects of new market entrants and the relocation of existing firms, addressing a gap in prior research. Second, this study examines how EID influences the implementation of other environmental regulatory tools. While previous research has analyzed the impact of EID on technology innovation through mechanisms such as human capital, foreign direct investment [30], political pressures, enforcement channels [29], innovation environment, investment, talent [31], green innovation environment, industrial structure [16], and corporate social responsibility [28], the interaction between different regulatory tools remains underexplored. To fill this gap, the study investigates how EID enhances public participation-based, command-control, and market-incentive regulations, analyzing whether EID can strengthen other regulatory tools to promote green technology innovation.

The subsequent content is structured as follows: The "Literature review and theoretical hypothesis" section reviews this study's related literature and theoretical hypotheses. The

"Methodology" section details the research design. The "Results and analysis" section presents the analysis and discussion of empirical results. The "Conclusions and policy implications" section concludes the article and proposes relevant policy recommendations.

## 2. Literature review and theoretical hypothesis

### 2.1 Literature review

There still needs to be more consensus within the academic community regarding the ability of environmental regulatory policies to stimulate enterprise technology innovation, resulting in two main viewpoints: incentive and inhibition. The "Porter Hypothesis" suggests that stringent environmental regulations raise production costs, compelling enterprises to engage in process and product innovations and ultimately enhancing their competitiveness [32]. Jaffe and Palmer (1997) further proposed the "narrow Porter Hypothesis," which posits that flexible and appropriate environmental regulatory policies can provide more significant incentives for innovation to enterprises [33]. However, neoclassical economics, driven by its advocacy for a free market economy and resistance to government intervention, argues that the "Porter Hypothesis" does not hold in certain circumstances [34]. In response to environmental regulatory policies, enterprises inevitably incur costs such as pollution emission fees and environmental pollution control investments, which compel them to reduce research and development investments [35, 36], thereby impeding improvements in technology innovation levels and production efficiency [37]. Consequently, numerous empirical studies have been conducted within academic circles to examine the innovative effects of various types of environmental regulations, including government-led environmental regulation [15, 38] and other forms of environmental regulations [18, 23, 38]. Overall, environmental regulations and technology innovation have a positive causal relationship. Although environmental regulations may impose significant "compliance costs" in the short term, they ultimately positively impact enterprise technology innovation in the long run.

With the expanding array of environmental regulatory tools, scholars have started exploring the impact of EID on enterprise innovation. EID primarily involves micro-entities (enterprises) and macro-entities (government departments), with the latter being the focus of this study. Existing research has demonstrated that EID can alleviate financing constraints, enhance green technology innovation, and boost business revenue [28, 39, 40]. However, some scholars contend that EID disrupts enterprises' path dependence, alters existing production habits, increases "compliance costs," reduces economic performance [24], and introduces new cost constraints [40], thereby dampening enterprises' willingness to innovate.

### 2.2 The direct effect of EID on enterprise green technology innovation

The existing research on the impact of environmental regulation on enterprise innovation forms the theoretical basis of this study. Neoclassical economic theory suggests that environmental regulations increase enterprises' "compliance costs," potentially hindering green innovation. In contrast, the "Porter Hypothesis" argues that well-designed regulations can encourage green technology innovation through an "innovation compensation" effect. Therefore, this study hypothesizes that EID positively influences enterprise green technology innovation.

Firstly, EID promotes enterprise green technology innovation through legitimacy pressures and reputation mechanisms. According to legitimacy theory, enterprises must follow social norms and environmental regulations to maintain their market legitimacy and reputation [41]. EID increases transparency by disclosing information about pollution and violations, subjecting enterprises to public and investor scrutiny. If the enterprise's actions deviate from

expectations, its legitimacy and market position may suffer, leading to potential losses in market share and investor trust [16]. As a result, enterprises need to adopt green technology that goes beyond regulatory requirements to build a responsible image [42].

Secondly, EID compels firms to engage in green technology innovation by internalizing externalities and maximizing long-term returns. Based on cost-benefit analysis theory, firms weigh investment costs against the direct and indirect benefits of innovation when making decisions regarding green technology investments. The transparency mechanism of EID directly transfers the costs of environmental pollution to enterprises, compelling them to assume greater environmental responsibilities [28]. In this context, enterprises consider the future costs of polluting behaviors and reassess the long-term benefits of green technology innovation. This reevaluation amplifies the perceived long-term gains from green technology innovation, encouraging firms to undertake green technology innovation activities.

Lastly, EID fosters industry-wide green technology advancement through competitive effects. By making environmental information transparent, EID shifts the focus of competition within industries from traditional factors like price and quality to environmental performance [43, 44]. Leading enterprises gain a "first-mover advantage" through green technology innovation, while other firms follow suit or pursue further innovation. This creates a demonstration and competitive effect within the industry, driving the overall upgrade and transition towards green technology.

Consequently, under the framework of EID, enterprises are motivated to engage in green technology innovation activities to a greater extent. This study presents the following hypothesis:

**Hypothesis 1:** EID can promote enterprise green technology innovation.

## 2.3 The indirect effect of EID on enterprise green technology innovation

EID enhances enterprise green technology innovation capacity by strengthening public participation-based environmental regulation. In market economies, information asymmetry often leads enterprises to prioritize short-term, non-sustainable friendly behaviors. EID improves transparency by providing stakeholders, including the public, with timely data on environmental performance and emissions. This transparency fosters greater public awareness and engagement in environmental oversight, reinforcing public participation in environmental governance [45]. The reduction in information asymmetry not only improves public awareness of enterprise environmental behavior but also grants the public greater oversight and participation rights, thereby giving them a stronger voice in environmental governance [46]. From a game theory perspective, the interactions among enterprises, the government, and the public can be viewed as a dynamic game. With pollution information made transparent, the public can exert pressure on firms through reporting, media exposure, and consumer boycotts, compelling firms to improve their environmental performance [45]. To mitigate compliance risks, protect their reputation, and avoid consumer backlash, firms often opt to improve their environmental performance through green technology innovation. This intrinsic motivation for innovation aligns with the "Porter Hypothesis", which suggests that firms adopting passive compliance strategies are unlikely to alleviate public scrutiny fully and may lose market share due to poor environmental performance [8]. Thus, enterprises are motivated to enhance environmental performance through innovation, maintaining a competitive edge.

EID enhances the enforcement of command-control environmental regulation, thereby promoting green technology innovation among enterprises. Command-control regulation rely on legal mandates and administrative measures to set pollution standards, ensuring

compliance through strict penalties and restrictions [47]. However, gaps between legislation and enforcement are present in China's current regulatory framework. Enterprises may circumvent environmental regulations through bribery and rent-seeking practices. EID improves transparency in environmental management processes, enabling greater oversight by the public and media over local governments and environmental agencies, thereby reducing opportunities for rent-seeking behaviors [48, 49]. According to public choice theory, government actions are often influenced by public and stakeholder interests. As societal demand for environmental protection grows, governments face pressure to implement stricter command-control policies to meet public expectations. Additionally, EID addresses information asymmetry between central and local governments. In China, environmental policies are formulated by the central government, while implementation is the responsibility of local authorities. However, local governments often possess more detailed information on enterprise emissions and compliance, creating challenges for the central government in monitoring local enforcement due to high information acquisition costs and time lags. By providing real-time environmental data, EID enables the central government to more accurately assess local environmental performance, thus strengthening accountability and incentive mechanisms and improving policy enforcement at the local level. Under stricter command-control regulations, enterprises face higher compliance requirements for pollution control, prompting technological upgrades in production processes, emission control, and waste management [31]. Based on cost-benefit analysis theory, firms confronted with significant fines or operational restrictions are more likely to weigh the short-term costs of compliance against the long-term benefits of technological upgrades. Consequently, firms are incentivized to adopt green technology innovations to mitigate future risks of fines and production disruptions.

EID strengthens the enforcement of market-incentive environmental regulation, thereby enhancing enterprise green technology innovation. Market-incentive regulation relies on price mechanisms and market signals to achieve effective pollution control. However, when enterprise environmental practices need more transparency, price signals may fail to accurately reflect accurate emission levels, limiting the effectiveness of market incentives. EID addresses this issue by providing transparent and timely environmental information, improving the accuracy of market signals. Investors and consumers can make informed economic decisions based on enterprises' environmental performance, such as engaging in green consumption or responsible investment. This strengthens the effectiveness of market incentives and increases demand for green technologies [50]. In response to these market incentives, enterprises must pursue technological innovation to gain a competitive edge, thereby accelerating the green innovation process. The internalization of externalities is a core objective of market-based environmental regulations. While enterprise pollution generates negative externalities, green technology innovation creates positive externalities. Traditional market mechanisms often fail to thoroughly account for these external costs and benefits. By disclosing enterprises' environmental data, EID enables society to assess external costs more accurately and pressure enterprises to assume environmental responsibilities. Greater information transparency allows market mechanisms, through the choices of consumers and investors, to better address the externalities of green technology innovation [8]. This market-incentive approach, facilitated by green consumption and responsible investment, further enhances the demand for green innovation. It encourages enterprises to address their environmental externalities and actively pursue green technological advancements.

Therefore, we put forward the following hypothesis:

**Hypothesis 2:** EID can enhance enterprise green technology innovation by strengthening public participation-based environmental regulation.

**Hypothesis 3:** EID can enhance enterprise green technology innovation by strengthening command-control environmental regulation.

**Hypothesis 4:** EID can enhance enterprise green technology innovation by strengthening market-incentive environmental regulation.

## 3. Methodology

### 3.1 Dates

This study focuses on Chinese A-share listed companies from 2003 to 2020. The COVID-19 pandemic has had a significant negative impact on normal development at all levels of the economy and society [51, 52]. We, therefore, defined the time frame of the study.

The green patent data of the listed companies used in this study is sourced from the China National Research Data Service (CNRDS). The specific processing steps are as follows: (1) The patent data of listed companies in CNRDS is used as the baseline, and the Intellectual Property Classification (IPC) codes of the patent data are obtained from the State Intellectual Property Office (SIPO) in China. (2) The acquired green patent data of listed companies are matched with the "International Patent Green Classification List" published by the World Intellectual Property Organization (WIPO) in 2010 to obtain the final data.In addition, the financial data of listed companies employed in this study is obtained from the China Stock Market and Accounting Research (CSMAR) database. Macro-level data is sourced from the China City Statistical Yearbook and the China Environmental Yearbook. Data on types of city newspapers were acquired from the China National Knowledge Infrastructure (CNKI) database, specifically the China Important Newspaper Full-text Database, and were manually retrieved. The following data processing steps were implemented: (1) Exclusion of companies in the financial sector. (2) Exclusion of ST and *ST type companies. (3) Elimination of samples with missing variables. (4) Trimming of continuous variables at the upper and lower 1% tails. Ultimately, a total of 3694 companies with 36076 valid N were obtained. The descriptive statistics of the variables are presented in Table 1.

**Table 1. Descriptive statistics.**

| Type | Variable | Symbol | N | Mean | SD | Min | Max |
|---|---|---|---|---|---|---|---|
| Dependent variable | The quantity of enterprise green technology innovation | GIT | 36076 | 0.605 | 1.011 | 0.000 | 7.030 |
| | The quality of enterprise green technology innovation | GII | 36076 | 0.249 | 0.640 | 0.000 | 6.753 |
| Independent variable | The DID dummy variable | PDID | 36076 | 0.738 | 0.440 | 0.000 | 1.000 |
| Mechanism variable | Public participation-based environmental regulation | PP | 36070 | -1.322 | 4.177 | -10.385 | 6.213 |
| | Command-control environmental regulation | CC | 36076 | 6.670 | 1.383 | 3.130 | 9.775 |
| | Market-incentive environmental regulation | MI | 36076 | 9.327 | 1.042 | 6.680 | 11.500 |
| Control variable | Firm size | Size | 36076 | 21.968 | 1.303 | 19.371 | 25.996 |
| | Leverage | Lev | 36076 | 0.440 | 0.213 | 0.057 | 0.972 |
| | Firm value | TobinQ | 36076 | 2.541 | 1.884 | 0.859 | 11.422 |
| | Firm age | Age | 36076 | 2.022 | 0.892 | 0.000 | 3.296 |
| | Owner concentration | Top1 | 36076 | 0.358 | 0.153 | 0.091 | 0.754 |
| | Cash holdings | Cash | 36076 | 0.197 | 0.147 | 0.011 | 0.702 |
| | Return on assets | Roa | 36076 | 0.036 | 0.065 | -0.278 | 0.195 |
| | Fixed asset ratio | Fix | 36076 | 0.229 | 0.172 | 0.002 | 0.730 |
| | Firm ownership | Soe | 36076 | 0.473 | 0.499 | 0.000 | 1.000 |

### 3.2 Models

Considering the introduction of the PITI in 2008 and the subsequent expansion of cities included in the assessment of PITI in 2013, this study utilizes a multi-period DID model to empirically examine the causal relationship between EID and enterprise green technology innovation. Following the methodology employed by Tan et al.(2022) [29], we establish the baseline model as follows:

$$GI_{it} = \beta_0 + \beta_1 PDID_{it} + \sum Control_{it} + \gamma_i + \delta_t + \mu_{it} \qquad (1)$$

Where i and t represent the firm and year, respectively, GI denotes the enterprise green technology innovation. PDID indicates EID, taking a value of 1 when the city where the enterprise is located discloses PITI and 0 otherwise. The estimated coefficient of PDID constitutes the primary focus of this study, discerning the net impact of EID on enterprise green technology innovation. Control encompasses other variables that influence enterprise green technology innovation. We also control firm-fixed effect ($\gamma$) and time-fixed effect ($\delta$) while employing firm-level cluster-robust standard error.

Furthermore, this study delves into how EID affects enterprise green technology innovation by examining public participation-based, command-control and market-incentive environmental regulation. Based on existing studies [53, 54], we construct the following model to investigate the mechanisms through which EID influences enterprise green technology innovation:

$$M_{it} = \theta_0 + \theta_1 PDID_{it} + \sum Control_{it} + \gamma_i + \delta_t + \mu_{it} \qquad (2)$$

$$GI_{it} = \phi_0 + \phi_1 M_{it} + \sum Control_{it} + \gamma_i + \delta_t + \mu_{it} \qquad (3)$$

Where M represents the mechanism variables, specifically public participation-based, command-control and market-incentive environmental regulation, the definitions of the remaining variables remain the same as in Eq (1).

### 3.3 Variables

**3.3.1 Dependent variabl.**   The dependent variable in this study is enterprise green technology innovation, encompassing the quantity of enterprise green technology innovation and the quality of enterprise green technology innovation. The study measures enterprise green technology innovation using granted green patent data and takes the natural logarithm after adding 1 to the data. This approach addresses the uncertainty in innovation input and the right-skewed distribution of patent data [55, 56]. In the robustness test, the granted patent data is substituted with application data to enhance the reliability of the findings. However, it is essential to note that there may be an issue of potential overestimation when measuring technology innovation using patent applications.

**3.3.2 Independent variable.**   The core independent variable in the multi-period DID model is PDID. If the enterprise is located in an area that is included in PITI cities in the current year and beyond, PDID = 1. If the enterprise is located in an area that does not publish PITI status, PDID = 0.

**3.3.3 Mechanism variable.**   The first mechanism variable is public participation-based environmental regulation. Public participation-based environmental regulation emphasizes the voluntary actions of market participants in controlling pollution. Existing studies typically use the number of public complaints and environmental proposals related to environmental issues as measurement indicators [16, 24, 57]. Therefore, this study uses the natural logarithm

of the number of environmental proposals in the city where the firm is located as a proxy for public participation-based environmental regulation.

The second mechanism variable is command-control environmental regulation. Command-control environmental regulation refers to government-imposed laws and environmental standards that limit corporate pollutant emissions to promote environmental protection. Prior literature commonly uses the number of environmental administrative penalties to measure the stringency of this regulation [47, 57]. Similarly, this study uses the natural logarithm of the number of environmental administrative penalties (plus one) in the city where the firm is located to capture the intensity of command-control regulation.

The third mechanism variable is market-incentive environmental regulation. Market-incentive environmental regulation leverages market pricing mechanisms to incentivize firms to assume environmental responsibility and consider pollution control in their production processes. Scholars frequently use the amount of pollution fees as a key indicator of this regulation [24, 58, 59]. Following this approach, we employ the natural logarithm of the pollution fees in the city where the firm is located to measure the strength of market-incentive environmental regulation.

It is worth noting that the data on proposal count, environmental administrative penalty cases, and pollution discharge fees are derived from the "*China Environmental Yearbook*" and are at the provincial level. Therefore, following the approach employed by Li et al.(2022) [16], this study applies city-level data obtained by weighting the provincial data based on the GDP proportion of cities within the province. For the number of environmental proposals, further weighting is conducted considering the population proportion of cities within the province to mitigate the influence of population factors on the number of proposals.

**3.3.4 Control variable.**   Control variables included in this study are based on existing research [15, 28, 60] that may impact enterprise green technology innovation. Firm size is measured by the natural logarithm of total assets. Leverage is measured by the ratio of total liabilities to total assets. Firm value is measured by the market value ratio to the cost of capital resetting. Firm age is measured by the natural logarithm of the year of listing plus 1. Owner concentration is measured by the percentage of shareholding held by the largest shareholder. Cash holdings are measured by the cash flow ratio generated from operating activities to year-end assets. Return on assets is measured by the net profit ratio to total assets. The fixed asset ratio is measured by the ratio of fixed assets to total assets. Firm ownership is a binary variable, assigning a value of 1 if the actual controller of a listed enterprise is a state-owned enterprise and 0 otherwise.

## 4. Results and analysis

### 4.1 Benchmark regression

The benchmark regression results are displayed in Table 2. Columns (1) and (2) present the regression outcomes without including control variables. The estimated coefficients for PDID exhibit significant positive effects at the 1% level, offering preliminary evidence for the beneficial impact of EID on green technology innovation in enterprises. Columns (3) and (4) present the regression outcomes with the inclusion of control variables. The findings in Column (3) demonstrate that PDID has a significantly positive effect on GIT at the 1% level, indicating a substantial increase in the overall quantity of enterprise green technology innovation. Column (4) results reveal that the estimated coefficients for PDID are all significantly positive at the 1% level, implying a positive influence on enterprise green technology innovation quality. As an emerging environmental regulatory tool, EID exhibits remarkable adaptability, fostering a remarkable "innovation compensation" effect for firms rather than a burden of "compliance

**Table 2. Benchmark regression results.**

| Variables | (1) | (2) | (3) | (4) |
|---|---|---|---|---|
| | GIT | GII | GIT | GII |
| PDID | 0.147*** | 0.097*** | 0.137*** | 0.092*** |
| | (0.041) | (0.028) | (0.041) | (0.028) |
| Size | | | 0.238*** | 0.116*** |
| | | | (0.019) | (0.013) |
| Lev | | | 0.063 | -0.011 |
| | | | (0.060) | (0.040) |
| TobinQ | | | 0.012*** | 0.006** |
| | | | (0.004) | (0.003) |
| Age | | | 0.007 | -0.007 |
| | | | (0.021) | (0.014) |
| Top1 | | | -0.286*** | -0.156** |
| | | | (0.106) | (0.073) |
| Cash | | | 0.262*** | 0.129*** |
| | | | (0.071) | (0.044) |
| Roa | | | -0.252*** | -0.196*** |
| | | | (0.081) | (0.055) |
| Fix | | | 0.075** | 0.038* |
| | | | (0.031) | (0.020) |
| Soe | | | -0.050 | -0.066* |
| | | | (0.054) | (0.037) |
| Constant | 0.036 | 0.002 | -4.945*** | -2.381*** |
| | (0.022) | (0.014) | (0.386) | (0.269) |
| Year fix effect | Yes | Yes | Yes | Yes |
| Firm fix effect | Yes | Yes | Yes | Yes |
| N | 36076 | 36076 | 36076 | 36076 |
| R-squared | 0.244 | 0.117 | 0.277 | 0.135 |

Robust standard errors clustered to the firm level are in parentheses

***, **, and * denote statistical significance at the 1%, 5%, and 10% levels, respectively.

costs," thereby effectively stimulating innovation. It positively impacts both the quantity and quality of enterprise green technology innovation, thus confirming hypothesis 1.

## 4.2 Parallel trend assumption and time trend

One assumption of the DID model is the parallel trend assumption, which asserts that the treatment and control groups should follow the same trend without policy intervention. Otherwise, the estimated results of the DID model may be biased. In this study, we employ the event analysis method to test the parallel trend assumption, drawing on the research conducted by Fang et al. (2019) [61]. Simultaneously, we utilize this method to examine the dynamic effects of EID.

The parallel trend test results for the baseline year, which is one year before the EID, are depicted in Fig 1. It is evident that prior to the EID, the estimated coefficients of PDID lack significance, suggesting no substantial disparity between the treatment and control groups. This signifies that the parallel trend assumption for the multi-period DID estimation holds valid. Furthermore, we delve deeper into investigating the dynamic effects of EID. The estimated PDID coefficients become statistically significant one year following the EID, indicating a

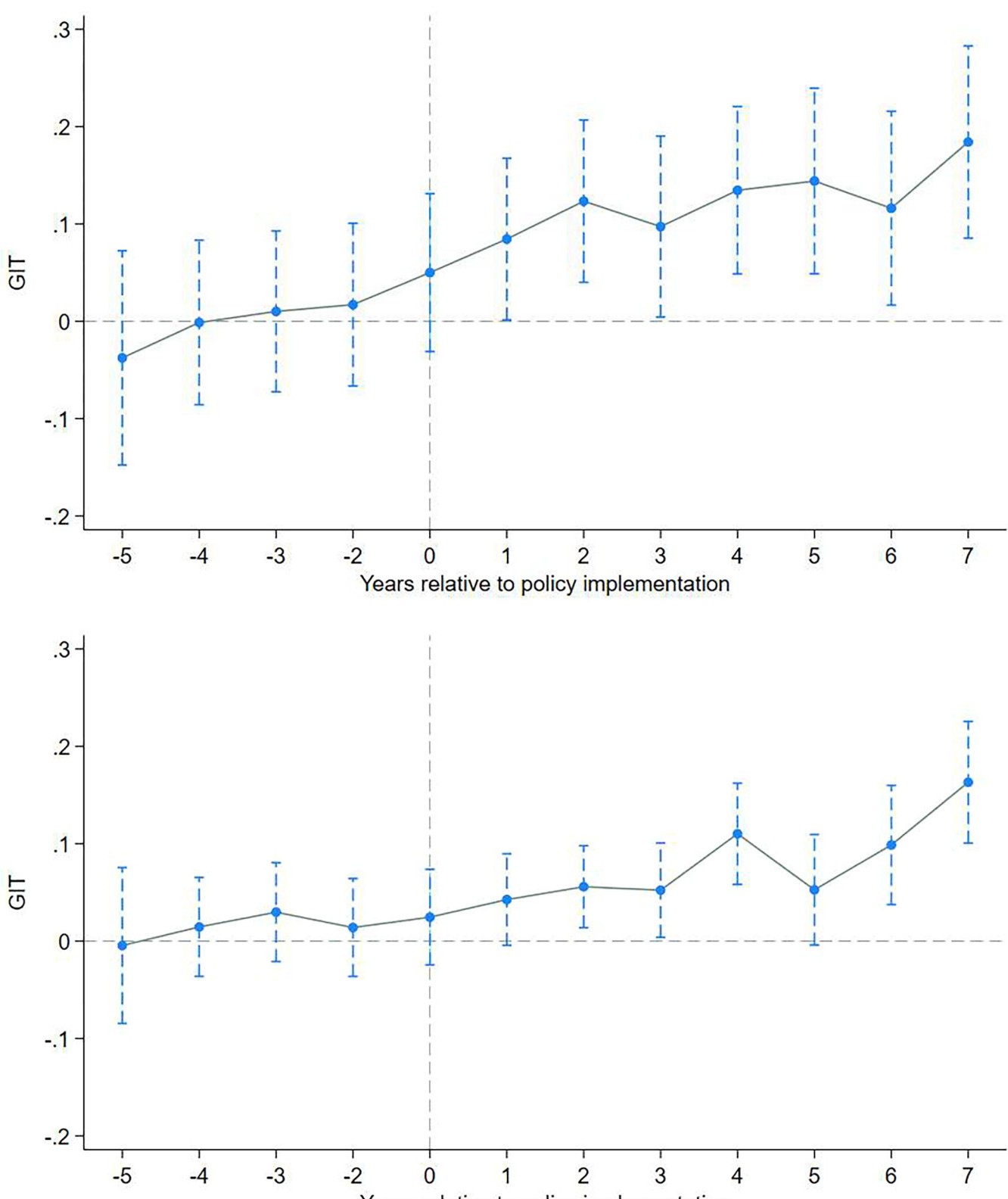

**Fig 1. Parallel trend assumption and time trend results.**

notable disparity in enterprise green technology innovation between the two sample groups. EID's promotion effect on enterprise green technology innovation demonstrates a delayed and enduring nature. Consequently, we can ascertain the reliability of the parallel trend test results and estimating the multi-period DID model satisfies the requirements.

### 4.3 Endogenous treatment and robustness tests

**4.3.1 Exclude self-selected samples.** Drawing upon the "Pollution Haven Hypothesis," enterprises consider disparities in environmental regulations when deciding on new locations or relocation. They tend to favor areas with lower levels of environmental regulation to minimize pollution control expenses. Once a city becomes part of the PITI, there is heightened pressure for EID. Enterprises burdened by high pollution control costs and limited technology innovation capabilities may opt to move away from such cities. Likewise, new enterprises also consider innovation capacities and pollution attributes when selecting the location. Consequently, the self-selection behavior of enterprises can introduce endogeneity issues, potentially undermining the credibility of the study's conclusions. To mitigate the influence of new entrants and relocation decisions, this study excludes data from newly established enterprises and samples of enterprise location changes after 2008 and then returned. As demonstrated in columns (1) and (2) of Table 3, the results remain statistically significant even after eliminating the interference caused by enterprises' voluntary location choices and relocations.

**4.3.2 Replace the interpreted variable.** There are also some issues with patent authorization data, such as long cycles and administrative intervention. Additionally, patents are already invested in the daily production of enterprises at the time of application and can more timely reflect the level of green technology innovation. Therefore, to alleviate potential measurement biases in the interpreted variables and address endogeneity issues in the research design, this study uses the number of green patent applications as an alternative indicator for green technology innovation and performs another regression. The results in columns (3) and (4) of Table 3 show that the conclusions remain robust, with the same signs and similar levels of significance as the baseline regression.

**4.3.3 PSM-DID.** This study further adopts the PSM-DID method to alleviate the endogenous problems caused by self-selection in cities evaluated by PITI. We use propensity score matching (PSM), with control variables as the main covariates, and employ a 1:4 nearest neighbor matching method with a caliper value of 0.05 to select research samples from the control group similar to the treatment group. Then, we estimate and overcome sample selection bias in the DID model. The results in columns (5) and (6) of Table 3 show that even after conducting PSM-DID regression, EID still significantly promotes both the quantity and quality of green technology innovation in enterprises, further enhancing the reliability of this study's findings.

**4.3.4 IV-2SLS.** In practice, factors such as the economic structure, cultural characteristics, and institutional environment of the city where an enterprise is situated can influence the evaluation of EID. These factors introduce endogeneity concerns and may impact the accuracy of the identification results. To address this issue, we draw on the studies by Shi et al. (2021) and Wang et al. (2022) [25, 62], and incorporate instrumental variables into the regression analysis of EID. As instrumental variables, we employ newspaper types (IV_News) and internet penetration rate (IV_Int). Newspaper types indicate media disclosure capabilities and information flow levels, representing public demand. A greater variety of newspaper types increases the likelihood of a city being included in the PITI list and meeting the relevance criteria. The internet penetration rate also satisfies the relevance requirements, as higher rates facilitate public access to environmental information and correspond to greater demand for EID. As for

**Table 3. Endogenous treatment and robustness test results.**

| Variables | (1) | (2) | (3) | (4) | (5) | (6) |
|---|---|---|---|---|---|---|
| | GIT | GII | GIT_Apply | GII_Apply | GIT | GII |
| PDID | 0.121*** | 0.092*** | 0.142*** | 0.140*** | 0.116*** | 0.081*** |
| | (0.046) | (0.032) | (0.045) | (0.038) | (0.044) | (0.030) |
| Size | 0.240*** | 0.118*** | 0.303*** | 0.251*** | 0.236*** | 0.115*** |
| | (0.020) | (0.014) | (0.020) | (0.018) | (0.019) | (0.012) |
| Lev | 0.041 | -0.019 | 0.005 | -0.004 | -0.004 | -0.016 |
| | (0.064) | (0.043) | (0.066) | (0.057) | (0.059) | (0.035) |
| TobinQ | 0.011*** | 0.005** | 0.009** | 0.012*** | 0.013*** | 0.007** |
| | (0.004) | (0.003) | (0.004) | (0.004) | (0.004) | (0.003) |
| Age | 0.002 | -0.013 | 0.033 | 0.001 | 0.015 | 0.004 |
| | (0.022) | (0.015) | (0.022) | (0.019) | (0.022) | (0.014) |
| Top1 | -0.181 | -0.112 | -0.334*** | -0.290*** | -0.280*** | -0.137** |
| | (0.116) | (0.082) | (0.112) | (0.099) | (0.105) | (0.068) |
| Cash | 0.225*** | 0.124*** | 0.206*** | 0.141** | -0.107* | -0.074* |
| | (0.075) | (0.047) | (0.078) | (0.066) | (0.060) | (0.038) |
| Roa | -0.293*** | -0.198*** | -0.155* | -0.152* | -0.298*** | -0.132** |
| | (0.086) | (0.059) | (0.090) | (0.078) | (0.086) | (0.056) |
| Fix | 0.087*** | 0.049** | 0.074** | 0.062** | 0.185*** | 0.095** |
| | (0.033) | (0.022) | (0.034) | (0.030) | (0.070) | (0.041) |
| Soe | -0.050 | -0.057 | -0.019 | -0.047 | 0.061** | 0.041** |
| | (0.057) | (0.039) | (0.059) | (0.051) | (0.029) | (0.018) |
| Constant | -5.030*** | -2.447*** | -6.204*** | -5.146*** | -4.854*** | -2.383*** |
| | (0.418) | (0.288) | (0.416) | (0.374) | (0.386) | (0.244) |
| Year fix effect | Yes | Yes | Yes | Yes | Yes | Yes |
| Firm fix effect | Yes | Yes | Yes | Yes | Yes | Yes |
| N | 33140 | 33140 | 36076 | 36076 | 24226 | 24226 |
| R-squared | 0.278 | 0.137 | 0.281 | 0.231 | 0.268 | 0.132 |

Robust standard errors clustered to the firm level are in parentheses

***, **, and * denote statistical significance at the 1%, 5%, and 10% levels, respectively.

exogenous factors, it becomes apparent that newspaper types do not directly drive green technology innovation in enterprises, and the internet penetration rate does not exhibit variations based on the level of green technology innovation in enterprises.

The estimation results of the IV-2SLS are presented in Table 4. Column (1) displays the first-stage regression results, indicating a significant positive correlation between newspaper types, internet penetration rate, and PDID. A higher number of newspaper types and a higher internet penetration rate are associated with more favorable conditions for EID. Columns (2) and (3) present the results of the second-stage regression, demonstrating that even after addressing potential endogeneity issues in the model, EID continues to promote green technology innovation in enterprises significantly. These findings further validate the robustness of the study's conclusions. Moreover, the selected instrumental variables, newspaper types, internet penetration rate, pass tests for weak instruments, over-identification, and identification robustness. Thus, employing newspaper types and internet penetration rate as instrumental variables can be deemed effective, with the regression results enhancing the credibility of the research conclusions.

**Table 4. Results of IV-2SLS.**

| Variables | (1) | (2) | (3) |
| --- | --- | --- | --- |
| | PDID | GIT | GII |
| IV_News | 0.015*** | | |
| | (0.001) | | |
| IV_Int | 0.019*** | | |
| | (0.002) | | |
| PDID | | 0.345* | 0.222* |
| | | (0.196) | (0.123) |
| Size | -0.002 | 0.239*** | 0.117*** |
| | (0.002) | (0.009) | (0.006) |
| Lev | 0.033*** | 0.052 | -0.017 |
| | (0.009) | (0.033) | (0.022) |
| TobinQ | -0.002*** | 0.012*** | 0.006*** |
| | (0.001) | (0.003) | (0.002) |
| Age | -0.013*** | 0.009 | -0.006 |
| | (0.003) | (0.012) | (0.008) |
| Top1 | 0.035** | -0.291*** | -0.160*** |
| | (0.014) | (0.054) | (0.037) |
| Cash | -0.018** | -0.047 | -0.064*** |
| | (0.008) | (0.036) | (0.024) |
| Roa | -0.006 | -0.248*** | -0.195*** |
| | (0.018) | (0.065) | (0.043) |
| Fix | -0.012 | 0.267*** | 0.130*** |
| | (0.011) | (0.039) | (0.025) |
| Soe | 0.019*** | 0.072*** | 0.036*** |
| | (0.006) | (0.018) | (0.012) |
| Constant | -0.018** | -0.047 | -0.064*** |
| | (0.008) | (0.036) | (0.024) |
| Year fix effect | Yes | Yes | Yes |
| Firm fix effect | Yes | Yes | Yes |
| N | 35632 | 35632 | 35632 |

Robust standard errors clustered to the firm level are in parentheses

***, **, and * denote statistical significance at the 1%, 5%, and 10% levels, respectively.

**4.3.5 Placebo test.** This study also conducts a placebo test to mitigate the impact of other unobservable factors on the conclusions [63]. We randomly assign 120 cities as the treatment group, while the remaining cities serve as the control group. A new interaction term is constructed, and the baseline regression is repeated 1000 times. Fig 2 illustrates the results of the placebo test, presenting a scatter plot depicting the kernel density distribution of the estimated coefficients of PDID following the 1000 regressions. It is evident that the estimated coefficients from the 1000 regressions are centered around zero, and the actual estimated value from the baseline regression does not fall within this distribution. The results of the placebo test align with expectations, indicating that the improvement in the level of green technology innovation in enterprises is indeed caused by EID, while ruling out interference from other unobserved factors.

**4.3.6 Control other policy interference.** In recent years, China has increasingly emphasized the construction of ecological civilization and innovation-driven development. It has

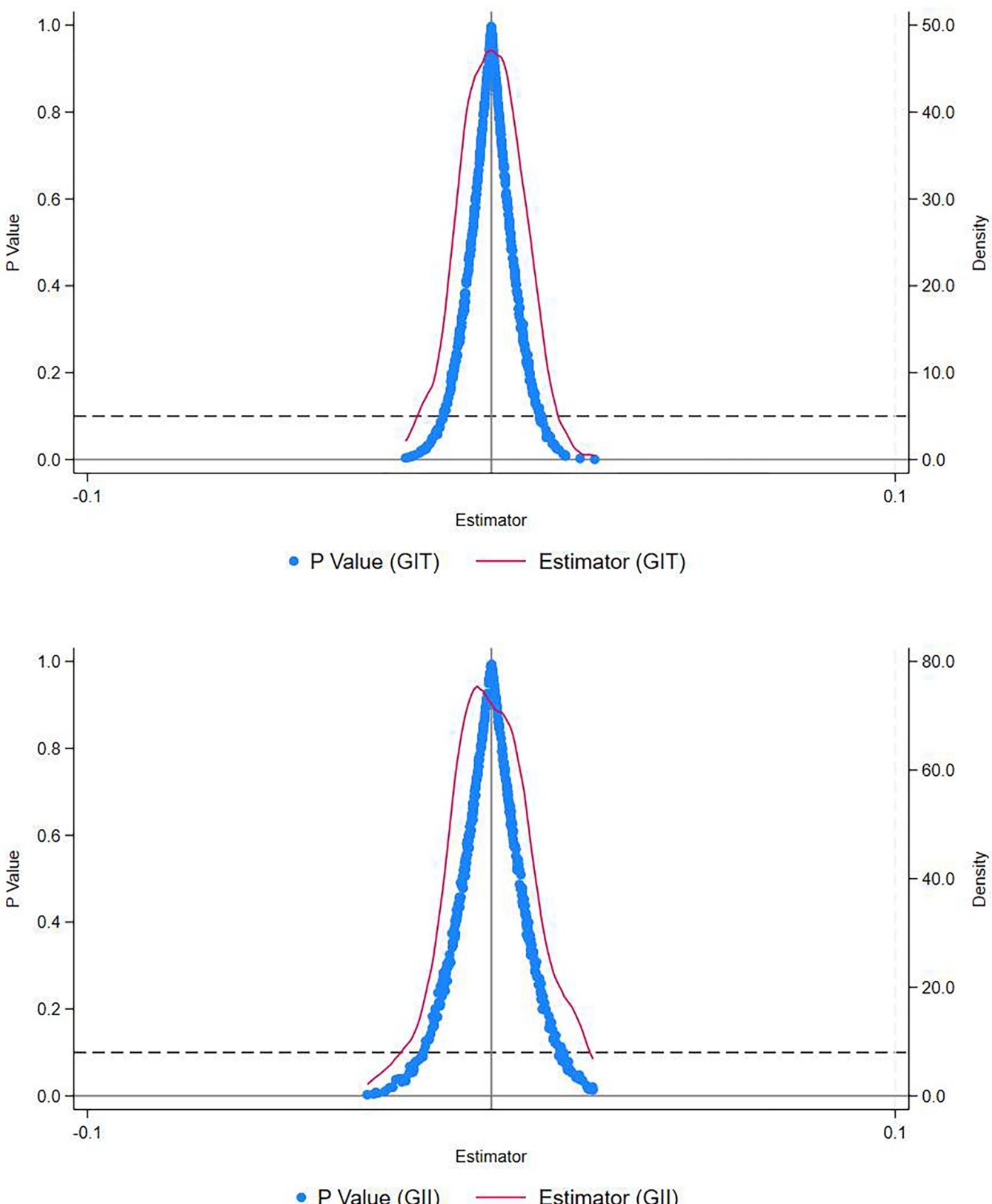

**Fig 2. Placebo test results.**

**Table 5. Results excluding other policy interferences.**

| Variables | (1) | (2) |
|---|---|---|
| | GIT | GII |
| PDID | 0.120*** | 0.068** |
| | (0.041) | (0.028) |
| CET | 0.011 | 0.045* |
| | (0.033) | (0.025) |
| IPMC | 0.042 | 0.032 |
| | (0.030) | (0.022) |
| LCC | 0.004 | 0.031** |
| | (0.023) | (0.016) |
| Constant | -4.950*** | -2.392*** |
| | (0.385) | (0.268) |
| Control variables fix effect | Yes | Yes |
| Year fix effect | Yes | Yes |
| Firm fix effect | Yes | Yes |
| N | 36076 | 36076 |
| R-squared | 0.277 | 0.136 |

Robust standard errors clustered to the firm level are in parentheses

***, **, and * denote statistical significance at the 1%, 5%, and 10% levels, respectively.

implemented a range of related policies that could have cross-impacts on the findings of this study. For instance, the Carbon Emission Trading Pilot Policy (CET) was initiated in 2011, while the National Intellectual Property Demonstration City Policy (IPMC) and the Low-carbon City Pilot Policy (LCC) were launched in 2012. To mitigate the influence of these policies, we have incorporated the double-difference terms of these policies into the baseline model. The regression results, presented in Table 5, demonstrate that even after accounting for the interference of other policies, the estimated coefficient of PDID remains significantly positive, affirming the robustness of the research findings.

## 4.4 Mechanism analysis

The results above indicate that EID enhances enterprise green technology innovation. This section examines the underlying mechanisms of this effect.

**4.4.1 Mechanism of public participation-based environmental regulation.** The results in Column (1) of Table 6 demonstrate a significantly positive estimated coefficient for PDID at the 1% level. EID plays a crucial role in promoting public participation-based environmental regulation, harnessing the public's potential in environmental protection, thus validating the previous theoretical analysis. Columns (2) and (3) reveal significantly positive estimated coefficients for public participation-based environmental governance, indicating that enhancing public participation-based environmental regulation can effectively drive enterprise green technology innovation. This confirms the existence of the mechanism behind public participation-based environmental regulation. EID amplifies the impact of public participation-based environmental regulation on enterprise green technology innovation, thereby supporting hypothesis 2. By providing the public with comprehensive pollution information, EID serves as a factual foundation for various stakeholders to pursue their environmental interests rationally and facilitates public engagement in environmental governance. Through social governance and market mechanisms, the public expresses their demands through actions like

Table 6. Mechanism analysis results of public participation-based environmental regulation.

| Variables | (1) | (2) | (3) |
|---|---|---|---|
|  | PP | GIT | GII |
| PDID | 0.720*** |  |  |
|  | (0.144) |  |  |
| PP |  | 0.015** | 0.012** |
|  |  | (0.007) | (0.005) |
| Constant | -2.823*** | -4.899*** | -2.344*** |
|  | (0.795) | (0.386) | (0.268) |
| Control variables fix effect | Yes | Yes | Yes |
| Year fix effect | Yes | Yes | Yes |
| Firm fix effect | Yes | Yes | Yes |
| N | 36070 | 36070 | 36070 |
| R-squared | 0.120 | 0.277 | 0.135 |

Robust standard errors clustered to the firm level are in parentheses

***, **, and * denote statistical significance at the 1%, 5%, and 10% levels, respectively.

"voting with their hands" and "voting with their money," rewarding or penalizing enterprises based on their pollution control and green innovation performance. Consequently, enterprises are compelled to accelerate their innovation in green technology.

**4.4.2 Mechanism of command-control environmental regulation.** Based on the results presented in Table 7, the estimated coefficients for PDID are significantly positive at the 1% level in column (1). This result suggests that EID reinforces command-control environmental regulation. In columns (2) and (3), the estimated coefficients for command-control environmental regulation are also significant, contributing to increased overall quantity and quality of green technology innovation in enterprises. Similarly, in column (4), EID demonstrates a strengthening effect on market-incentive environmental regulation. Furthermore, market-

Table 7. Mechanism analysis results of command-control environmental regulation.

| Variables | (1) | (2) | (3) | (4) | (5) | (6) |
|---|---|---|---|---|---|---|
|  | CC | GIT | GII | MI | GIT | GII |
| PDID | 0.455*** |  |  | 0.177*** |  |  |
|  | (0.062) |  |  | (0.043) |  |  |
| CC |  | 0.025* | 0.030*** |  |  |  |
|  |  | (0.014) | (0.010) |  |  |  |
| MI |  |  |  |  | 0.057*** | 0.054*** |
|  |  |  |  |  | (0.014) | (0.011) |
| Constant | 6.487*** | -5.102*** | -2.574*** | 8.259*** | -5.414*** | -2.822*** |
|  | (0.328) | (0.392) | (0.275) | (0.285) | (0.402) | (0.286) |
| Control variables fix effect | Yes | Yes | Yes | Yes | Yes | Yes |
| Year fix effect | Yes | Yes | Yes | Yes | Yes | Yes |
| Firm fix effect | Yes | Yes | Yes | Yes | Yes | Yes |
| N | 36076 | 36076 | 36076 | 36076 | 36076 | 36076 |
| R-squared | 0.131 | 0.277 | 0.136 | 0.244 | 0.278 | 0.138 |

Robust standard errors clustered to the firm level are in parentheses

***, **, and * denote statistical significance at the 1%, 5%, and 10% levels, respectively.

**Table 8. Mechanism analysis results of market-incentive environmental regulation.**

| Variables | (1) | (2) | (3) |
|---|---|---|---|
| | MI | GIT | GII |
| PDID | 0.177*** | | |
| | (0.043) | | |
| MI | | 0.057*** | 0.054*** |
| | | (0.014) | (0.011) |
| Constant | 8.259*** | -5.414*** | -2.822*** |
| | (0.285) | (0.402) | (0.286) |
| Control variables fix effect | Yes | Yes | Yes |
| Year fix effect | Yes | Yes | Yes |
| Firm fix effect | Yes | Yes | Yes |
| N | 36076 | 36076 | 36076 |
| R-squared | 0.244 | 0.278 | 0.138 |

Robust standard errors clustered to the firm level are in parentheses

***, **, and * denote statistical significance at the 1%, 5%, and 10% levels, respectively.

incentive environmental regulation significantly enhances both the total quantity and quality of enterprise green technology innovation, as evidenced by the results in columns (5) and (6). In summary, EID plays a crucial role in augmenting both command-control and market-incentive government-led environmental regulations, thus influencing enterprise green technology innovation. These findings align with the previous theoretical analysis and provide empirical support for hypothesis 3. One plausible explanation for these results is that EID intensifies the oversight exerted by higher-level governments and society, compelling local governments to implement and reinforce command-control environmental regulation. The imposition of environmental penalties and pollution fees increases the cost of pollution for enterprises, impacting their reputation and legitimacy and consequently motivating them to transition from short-term reactive behaviors under environmental governance to long-term developmental decisions. Ultimately, this drives improvements in enterprise green technology innovation.

**4.4.3 Mechanism of market-incentive environmental regulation.** The results in column (1) of Table 8 show that the estimated coefficients of PDID are significantly positive, proving a significant reinforcing effect of EID on market-incentive environmental regulation. The results in columns (2) and (3) show that market-incentive environmental regulation significantly increases the total quantity and quality of enterprise green technology innovation. The above results reveal that EID can strengthen the market incentive-based environmental regulation, which in turn enhances the level of green technology innovation of enterprises and verifies Hypothesis 4. The reason is that EID strengthens the information mechanism, which is conducive to improving the transparency of enterprises' environmental behavior, and it also enhances the role of market-incentive environmental regulation so that enterprises are forced to consider green technology innovation activities in production and operation. Moreover, the promotion effect of market incentive-based environmental regulation on enterprises' green technology innovation has been generally confirmed.

## 4.5 Heterogeneity analysis

Regional variations may influence the impact of EID on enterprise green technology innovation in marketization levels and resource endowments. To examine this heterogeneity, this

**Table 9. Heterogeneity analysis results.**

| Variables | (1) | (2) | (3) | (4) |
|---|---|---|---|---|
| | GIT | GII | GIT | GII |
| PDID | 0.050 | 0.004 | 0.123*** | 0.094*** |
| | (0.049) | (0.029) | (0.042) | (0.029) |
| Market | -0.040 | -0.071*** | | |
| | (0.036) | (0.021) | | |
| PDID×Market | 0.103*** | 0.103*** | | |
| | (0.037) | (0.022) | | |
| Resource | | | 0.007 | 0.057 |
| | | | (0.108) | (0.067) |
| PDID×Resource | | | 0.150** | 0.008 |
| | | | (0.072) | (0.044) |
| Constant | -4.937*** | -2.351*** | -4.945*** | -2.393*** |
| | (0.384) | (0.267) | (0.387) | (0.270) |
| Control variables fix effect | Yes | Yes | Yes | Yes |
| Year fix effect | Yes | Yes | Yes | Yes |
| Firm fix effect | Yes | Yes | Yes | Yes |
| N | 36076 | 36076 | 36076 | 36076 |
| R-squared | 0.278 | 0.136 | 0.278 | 0.135 |

Robust standard errors clustered to the firm level are in parentheses

***, **, and * denote statistical significance at the 1%, 5%, and 10% levels, respectively.

study incorporates dummy variables for city marketization level (Market) and resource endowments (Resource) into the baseline model (1) as well as interaction terms with PDID.

**4.5.1 Heterogeneity of marketization level.** Marketization refers to the enhancement of resource allocation efficiency and the development of a market economy. It plays a crucial role in facilitating the participation of non-governmental organizations and the public in environmental governance and, therefore, serves as an external driving force for enterprise green technology innovation. Given the significant economic disparities across regions in China, the level of marketization varies, leading to disparities in the effectiveness of EID. Building upon the research conducted by Tang et al. (2020) [64], this study calculates the marketization level of the city where the enterprise is located and introduces a dummy variable (Market) to represent this level. If the city's marketization level exceeds the average level, the Market is assigned a value of 1; otherwise, it is assigned a value of 0.

The regression results regarding marketization level heterogeneity are presented in columns (1) and (2) of Table 9. These results indicate that the estimated coefficient of PDID×Market is significantly positive, implying that the impact of EID on enterprise green technology innovation is more pronounced for samples located in highly marketized regions. This finding can be attributed to the fact that higher levels of marketization enable the market mechanism to play a more influential role. Moreover, the public tends to prefer green and environmentally friendly products and services in such regions, which stimulates enterprises' inclination to innovate in green technology. Simultaneously, a higher level of marketization effectively mitigates price distortions and misallocations of innovation factors, fostering an environment conducive to developing green innovation activities in enterprises.

**4.5.2 Heterogeneity of resource endowment.** Resource-based cities have experienced rapid development by relying on natural resources. However, an excessive dependency on these resources has led to a detrimental path dependency, resulting in a lack of motivation and

capacity for innovative development, ultimately leading to a "resource trap." In 2013, the State Council of China issued the *National Sustainable Development Plan for Resource-based Cities*, identifying 262 resource-based cities, including 126 administrative regions at the prefecture level. This study matches enterprise data with the corresponding resource-based cities. Creating a dummy variable (Resource) for resource-based cities and assigning the value of 1 if Enterprises successfully matched are categorized as resource-based city samples. Otherwise, it is 0.

Columns (3) and (4) in Table 9 present the results of the heterogeneity regression analysis based on resource endowments. The results reveal that the estimated coefficients of PDID×Resource are positive for both GIT and GII, with statistical significance observed only for GIT, which means that the impact of EID on enterprise green technology innovation is more pronounced in samples characterized by substantial resource endowments. It may be attributed to the abundant natural resources and energy endowments in cities with solid resource capabilities. In such cities, enterprises primarily engage in resource extraction and processing industries, which often generate significant environmental pollution. EID compels governments in resource-rich cities to raise environmental standards, thereby increasing external pressures faced by enterprises and driving them to intensify their efforts in enterprise green technology innovation. To grant a patent, inventive green patents signify more complex technological content and longer commercialization cycles. Due to time and resource constraints, enterprises in resource-based industries prioritize practical patents offering short-term benefits. As a result, EID significantly impacts the quantity of enterprise green technology innovation but may not necessarily affect its quality.

## 5. Conclusions and policy implications

### 5.1 Discussion

Green technology logical innovation offers dual benefits in economic development and environmental governance and is essential to achieving sustainable development [5, 7]. EID possesses significant potential to enhance green technology innovation by disclosing information on the state of urban environmental pollution. However, existing studies have paid limited attention to how city-level EID affects enterprise green innovation and need more empirical evidence. Previous studies have highlighted how firm-level EID affects corporate green technology innovation [65, 66] and have also explored how city-level EID affects regional green technology innovation [16, 61]. Our study broadens the understanding of the city-level EID innovation effect to encompass the firm level, confirming that EID is equally essential for enterprise green technology innovation. This result supports previous studies [28, 36]. Unlike some previous studies, it was found that the technology innovation effect from EID not only affects the quantity but also significantly affects the quality of innovation. Moreover, the study sample was expanded from one industry to all industries, significantly increasing the EID policy's scope. EID is a flexible environmental regulation policy that provides a buffer for firms to adjust their internal resource allocation and cope with environmental regulations, serving as an effective tool to improve green technology innovation.

In addition, existing research on the internal mechanism of EID in promoting enterprises' green technology innovation has yet to be fully explored. Lu and Li (2023) analyzed the regulating mechanism of digital transformation but did not analyze the channels of EID's impact on green technology innovation [66]. Other scholars have examined the influence mechanisms from the perspectives of information asymmetry, financing constraints [67], and corporate social responsibility [28]. However, most of the above mechanisms have been explored from within the firm, and few studies have analyzed them from the perspective of the firm's external

system, especially the heterogeneous environmental regulatory instruments. In this study, it is believed that EID can form a multi-party environmental governance system. Specifically, EID has a noticeable promotional effect on public participation-based, command-control, and market-incentive environmental regulation. By forming the linkage effect of diversified environmental regulation policies, external environmental pressure on enterprises is enhanced, forcing them to turn to green technology innovation to adapt to the new environmental governance system.

## 5.2 Conclusions

Based on the data of Chinese A-share listed companies from 2003 to 2020, this study leverages the PITI as a quasi-natural experiment for EID. It employs a multi-period DID model to investigate how EID can effectively foster enterprise green technology innovation. The findings unequivocally demonstrate that EID significantly enhances enterprise green technology innovation, both in terms of innovation quantity and quality. Robustness checks and a series of endogeneity tests further affirm the reliability of the conclusion, showcasing the strengthening impact of EID on green technology innovation in companies over time and suggesting the presence of dynamic effects. Moreover, this study delves into the potential of EID to stimulate the involvement of diverse stakeholders in environmental governance, encompassing both public participation-based and government-led environmental regulation. Mechanism analysis validates that EID bolsters command-control and market-incentive government-led environmental regulations while reinforcing public participation-based environmental regulation, ultimately fostering an elevated enterprise green technology innovation. Additionally, heterogeneity analysis reveals that the efficacy of EID is contingent on marketization levels and resource endowments. Notably, in samples characterized by high marketization and substantial resource endowment, the influence of EID on driving enterprise green technology innovation is particularly pronounced.

## 5.3 Policy implications

This study offers valuable insights for countries with imperfect environmental governance systems, aiding them in enhancing public participation-based environmental regulation through EID while constraining the implementation of government-led environmental regulation. Moreover, it helps establish an environmental pollution governance system that fosters enterprise green technology innovation. The study proposes the following recommendations:

(1) In light of growing ecological constraints, the EID policy should be applied judiciously, ensuring the authenticity and timeliness of the disclosed information while continuously expanding the scope and content of EID. Based on this study's findings, which demonstrate the positive outcomes of EID in pollution control and promoting enterprise green technology innovation, it is crucial to scientifically and reasonably expand and enhance EID. Therefore, it can be achieved through summarizing existing experiences, improving the quality of EID, and optimizing information accessibility. Such efforts will facilitate the orderly implementation of EID in more cities.

(2) We must construct an environmental governance system by leveraging EID, optimizing the combination of environmental regulatory tools, and establishing a diversified governance framework involving multiple stakeholders. On the one hand, EID serves as the foundation for the participation of various social entities in environmental governance, and the involvement of these entities is essential for the long-term development of EID. Government should provide Active guidance to encourage and engage public participation, non-governmental organizations (NGOs), and other entities in the environmental governance system. Strengthening

environmental awareness and utilizing the opportunities presented by EID will enhance social supervision. On the other hand, for EID to be adequate, it must be supported by appropriate policies and institutional frameworks. Assessing the effectiveness of local governments' environmental governance efforts should be improved, utilizing the role of local governments as a warning and guiding force in environmental governance. Specifically, This includes transmitting stable and positive signals regarding green and low-carbon transformation and environmental governance to the financial and product markets, enhancing enterprises' economic and political motivation to improve production technology and reduce pollution and emissions.

(3) Considering the diversity of local institutional environments and resource characteristics, external interventions must be strengthened to guide enterprise green transformation rationally. While addressing the externalities of environmental governance, the government should promote the positive incentive effects of EID on microeconomic innovation activities by enhancing marketization and achieving coordination and complementarity between market institutions and environmental institutions. Simultaneously, efforts should be accelerated to promote mature green technologies that offer enterprises both environmental and economic benefits. The main ways include developing policy mechanisms and market environments incentivizing enterprise green technology innovation. In regions with resource endowment advantages, long-term development plans should be formulated promptly to avoid the resource curse through green, low-carbon, and innovation-driven development. The government can achieve these by optimizing the regional industrial structure, diversifying industrial types, and introducing new technologies and industries to prevent developmental decline after resource depletion.

## 5.4 limitations and future research directions

There are several limitations to this study, along with new directions for future research. Firstly, these findings are derived solely from listed companies and do not consider the impact of EID on green technology innovation in SMEs. Listed companies tend to be better able to cope with environmental regulatory changes, possess greater risk resilience and R&D capabilities, and place greater emphasis on corporate reputation and image. Therefore, the impact of EID on SMEs may be overestimated or negligible. Due to the large gap between SMEs and listed companies, SMEs' actions in the face of environmental regulatory policies such as EID will be adjusted accordingly. Therefore, future research should further focus on how SMEs respond to stronger external environmental regulations. This includes examining which methods, such as production reduction, relocation, and innovation, will be chosen by SMEs to better adapt to the institutional environment. Second, due to data limitations, this study needs to explore the participation of media and institutional investors in environmental governance and limit the potential impact of EID on other environmental regulatory tools to those based on public participation and government-led regulation. When exploring other environmental policies, scholars can build a synergistic environmental governance mechanism that involves multiple actors such as the government, the public, the media, and institutions and investigate the importance and differences in the influence of each stakeholder on corporate green technology innovation.

## Author Contributions

**Conceptualization:** Weigang Ma.

**Data curation:** Xingqi Wang.

**Formal analysis:** Xingqi Wang.

**Funding acquisition:** Weigang Ma.

**Investigation:** Ji Li.

**Methodology:** Xingqi Wang.

**Project administration:** Xingqi Wang.

**Resources:** Xingqi Wang.

**Software:** Xingqi Wang.

**Supervision:** Weigang Ma.

**Validation:** Weigang Ma.

**Visualization:** Ji Li.

**Writing – original draft:** Xingqi Wang.

**Writing – review & editing:** Kun Zhang, Weigang Ma.

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
