## [Decision Letter · Decision Letter 0]

23 Nov 2023

PONE-D-23-33221Does environmental information disclosure promote enterprise green technology innovation?PLOS ONE

Dear Dr. Wang,

Thank you for submitting your manuscript to PLOS ONE. After careful consideration, we feel that it has merit but does not fully meet PLOS ONE’s publication criteria as it currently stands. Therefore, we invite you to submit a revised version of the manuscript that addresses the points raised during the review process.

We look forward to receiving your revised manuscript.

Kind regards,

Syed Usman Qadri, PhD

Academic Editor

PLOS ONE

Journal Requirements:

4. Please ensure that you include a title page within your main document. You should list all authors and all affiliations as per our author instructions and clearly indicate the corresponding author.

Additional Editor Comments:

Manuscript is well written. However, Author must incorporate following comments along with reviewer suggestions: 1. English proof read required 2. Introduction should be revised and address research gap deeply.3. Discuss methodology section more focused and explain varriable.4 reference should be updated.

Reviewers' comments:

Reviewer's Responses to Questions

Comments to the Author

1. Is the manuscript technically sound, and do the data support the conclusions?

Reviewer #1: Yes

Reviewer #2: Yes

2. Has the statistical analysis been performed appropriately and rigorously? 

Reviewer #1: Yes

Reviewer #2: Yes

3. Have the authors made all data underlying the findings in their manuscript fully available?

Reviewer #1: No

Reviewer #2: Yes

4. Is the manuscript presented in an intelligible fashion and written in standard English?

Reviewer #1: No

Reviewer #2: Yes

5. Review Comments to the Author

Reviewer #1: Review Report

Summary

The manuscript's topic is interesting, but the study has several drawbacks. The authors are encouraged to implement the revisions and suggestions provided below to enhance the quality of the paper.

Major Issues

1.The abstract should be in structured form and contain the main crux of the study.

2.The author has not provided an adequate description of the potential effects of green technology on various sectors or industries.

3.The author has employed excessive line spacing, leading to an increased page count. It is advisable to eliminate the additional line spacing to optimize the document's formatting and page count.

4.The does not provide the future direction of the study which is essential for any research study.

5.Please make the concluding section shorter and a little bit longer.

6.Please compare the findings with those of the others.

7.In the final manuscript, I've observed that a majority of the Figures and Tables are not cited in the main text. Please ensure that these are properly cited in the DOC format under the Main Manuscript section on Phenom to align with best practices.

8.On which basis the author has select the companies from the China Stock Market and also provide complete source of data collection.

Minor Issues

9.References should be alphabetically ordered for clarity and alignment with journal guidelines.

10.It has come to my attention that the author has overlooked including DOI numbers for several references. To enhance the comprehensiveness of the citation information, please ensure that the DOI numbers are added to these references.

11.Please make the appropriate adjustments to the references' in-text citation styles.

12.The author should update the manuscript with more recent references and also include the following additional references.

https://doi.org/10.3389/fpubh.2022.1055406

https://doi.org/10.1155/2023/9536571

https://doi.org/10.3389/fpsyg.2022.953454

https://doi.org/10.3389/fenvs.2023.1074713

https://doi.org/10.3389/fenvs.2023.1067531

13.All of the references in the final list should be written in the same style.

Other Comments

14.The paper's presentation might use some improvement. Please correct the writing and spacing errors.

15.The paper contains a number of grammatical errors. Please fix these, especially in Introduction, Literature Review and Conclusion.

16. Please include in-text citations for all the tables and figures correctly and upload the final updated manuscript.

17.Provide clearer pictures about all Figures. Further, need to explain the meaning of the existence of figures.

18.The discussion part of the research is missing. Make a heading only "Discussion" and then write the discussion align with your study results. Authors need to search the latest literature and write the discussion section.

Reviewer #2: Thank you for inviting me to review this work. I thoroughly reviewed this work and found it interesting and meaningful with merits. This study valuable insights for countries with imperfect environmental governance system. Based on study design, model, and analysis, I would recommend it for publication with minor language editing.

Best of Luck

6. PLOS authors have the option to publish the peer review history of their article (what does this mean?). If published, this will include your full peer review and any attached files.

Do you want your identity to be public for this peer review? For information about this choice, including consent withdrawal, please see our Privacy Policy.

Reviewer #1: No

Reviewer #2: No

---

## [Author Response · Author response to Decision Letter 0]

19 Jun 2024

We are very grateful for the valuable feedback and insightful comments provided on our manuscript. We appreciate the time and effort you have invested in reviewing our work.

We have carefully considered all the comments and suggestions and have made substantial revisions to the manuscript accordingly. Below, we provide detailed responses to each point raised.

Editor Comments:

1. Comment: "Manuscript is well written. However, Author must incorporate following comments along with reviewer suggestions: 1. English proof read required 2. Introduction should be revised and address research gap deeply. 3. Discuss methodology section more focused and explain variable. 4. Reference should be updated."

Response:

1. We have thoroughly proofread the revised manuscript in English and corrected grammatical errors.

2. In the introduction section, we have added relevant studies on green technology innovation to further highlight the importance of the research problem addressed in this paper.

3. We have expanded and optimized the methodology section, providing a more detailed explanation of the explanatory variables.

4. We have reviewed and updated the references to include the most recent and relevant literature.

Reviewer 1 Comments:

1. Comment: "The abstract should be in structured form and contain the main crux of the study."

Response: We appreciate this comment and have restructured the abstract to clearly define the research question and its significance.

2. Comment: "The author has not provided an adequate description of the potential effects of green technology on various sectors or industries."

Response: We have expanded the introduction to include information on the impact of green technology innovation across various sectors, emphasizing its importance and necessity.

3. Comment: "The author has employed excessive line spacing, leading to an increased page count. It is advisable to eliminate the additional line spacing to optimize the document's formatting and page count."

Response: We have revised the document to single-spacing, addressing the issue of excessive line spacing.

4. Comment: "The study does not provide future directions which are essential for any research study."

Response: Considering the study's limitations, we have included guidance for future research directions in Section 5.

5. Comment: "Please make the concluding section shorter and a little bit longer."

Response: We have expanded the conclusion section to provide a more comprehensive summary of our findings.

6. Comment: "Please compare the findings with those of other studies."

Response: In the discussion section of Section 5, we have compared our findings with previous research, highlighting the uniqueness and contributions of our study.

7. Comment: "In the final manuscript, a majority of the Figures and Tables are not cited in the main text. Please ensure that these are properly cited in the DOC format under the Main Manuscript section on Phenom to align with best practices."

Response: Figures have been placed within the main text of the manuscript and properly cited.

8. Comment: "On what basis were the companies from the China Stock Market selected? Please provide the complete source of data collection."

Response: We have outlined the criteria for selecting Chinese stock market firms and explained the basis for the study's timeframe.

9. Comment: "References should be alphabetically ordered for clarity and alignment with journal guidelines."

Response: The format and arrangement of references have been revised according to PLOS ONE journal guidelines.

10. Comment: "The author has overlooked including DOI numbers for several references. Please ensure that DOI numbers are added to enhance the comprehensiveness of the citation information."

Response: DOI numbers have been added to all references in the revised manuscript.

11. Comment: "Please make the appropriate adjustments to the references' in-text citation styles."

Response: The citation format of references has been standardized.

12. Comment: "The author should update the manuscript with more recent references and include the following additional references."

Response: We have updated the references. Based on the reviewers' suggestions, we have cited the following documents: doi:10.1155/2023/9536571 and doi:10.3389/fpsyg.2022.953454. Other references have also been updated in the revised manuscript.

13. Comment: "All of the references in the final list should be written in the same style."

Response: All references in the final list have been written in the same style.

14. Comment: "The paper's presentation might need some improvement. Please correct the writing and spacing errors."

Response: Writing and spacing errors have been corrected.

15. Comment: "The paper contains several grammatical errors. Please fix these, especially in the Introduction, Literature Review, and Conclusion."

Response: Grammatical errors have been corrected in the Introduction, Literature Review, and Conclusion sections.

16. Comment: "Please include in-text citations for all the tables and figures correctly and upload the final updated manuscript."

Response: All tables and figures have been correctly cited in the final manuscript.

17. Comment: "Provide clearer pictures about all Figures. Further, need to explain the meaning of the existence of figures."

Response: We provided clear pictures of all the figures and explained their significance in the context of the content.

18. Comment: "The discussion part of the research is missing. Make a heading only 'Discussion' and then write the discussion align with your study results. Authors need to search the latest literature and write the discussion section."

Response: We have added a discussion section to analyze relevant studies by other scholars and to highlight the significance of this study.

Reviewer 2 Comments:

1. Comment: "Thank you for inviting me to review this work. I thoroughly reviewed this work and found it interesting and meaningful with merits. This study provides valuable insights for countries with imperfect environmental governance systems. Based on study design, model, and analysis, I would recommend it for publication with minor language editing."

Response: We are grateful for your positive feedback and recommendation. We have made minor language edits throughout the manuscript to improve clarity and readability.

---

## [Decision Letter · Decision Letter 1]

10 Sep 2024

PONE-D-23-33221R1Does environmental information disclosure promote enterprise green technology innovation?PLOS ONE

Dear Dr. Wang,

Thank you for submitting your manuscript to PLOS ONE. After careful consideration, we feel that it has merit but does not fully meet PLOS ONE’s publication criteria as it currently stands. Therefore, we invite you to submit a revised version of the manuscript that addresses the points raised during the review process.

We look forward to receiving your revised manuscript.

Kind regards,

Zhaoyang Zhao

Guest Editor

PLOS ONE

Reviewers' comments:

Reviewer's Responses to Questions

**Comments to the Author**

1. If the authors have adequately addressed your comments raised in a previous round of review and you feel that this manuscript is now acceptable for publication, you may indicate that here to bypass the “Comments to the Author” section, enter your conflict of interest statement in the “Confidential to Editor” section, and submit your "Accept" recommendation.

Reviewer #1: All comments have been addressed

Reviewer #3: All comments have been addressed

2. Is the manuscript technically sound, and do the data support the conclusions?

Reviewer #1: Yes

Reviewer #3: Partly

3. Has the statistical analysis been performed appropriately and rigorously? 

Reviewer #1: Yes

Reviewer #3: No

4. Have the authors made all data underlying the findings in their manuscript fully available?

Reviewer #1: Yes

Reviewer #3: Yes

5. Is the manuscript presented in an intelligible fashion and written in standard English?

Reviewer #1: Yes

Reviewer #3: No

6. Review Comments to the Author

Reviewer #1: Author incorporate all the changes highlighted in comments. The author incorporate the changes and fix these in effective manner. Now the time to publish the author work I highly recommend it.

Reviewer #3: The innovation point of this article needs to be more precise. There is already some clear literature on the correlation between PITI and green technology innovation. The author needs to seek their unique features humbly.

For the multi period DID model, the construction of Model 1 in this article is incorrect and requires significant adjustments. As this article uses multi period DID, it's crucial to provide rigorous evidence based on the current research progress of methods. The current recognition results are not reliable because only the case of classical DID is considered.

Why does local governments' disclosure of environmental information directly lead to green technology innovation in enterprises? Whether in the theoretical hypothesis discussion in this article or in the empirical strategies later, this theory and logic have not provided satisfactory answers. According to relevant literature, the vast majority of enterprises are completely unaware of this PITI. If this article believes that PITI can affect the quality of green innovation, what is the specific path?

In 3.3.3 Mechanism variable, both government-led environmental regulation and public participation-based environmental regulation are considered instrumental variables, which conflicts with the title.

What is the basis for selecting public participation-based environmental regulation and government-led environmental regulation in the 4.4 Mechanism Analysis section, and which category should the PITI in this article belong to? We should not be enthusiastic about creating concepts but focus on rigorous argumentation.

The citation of references in this article is inaccurate, and the language expression requires professional polishing to make the article more readable.

7. PLOS authors have the option to publish the peer review history of their article (what does this mean?). If published, this will include your full peer review and any attached files.

Reviewer #1: **Yes: **Mohsin Raza

Reviewer #3: No

---

## [Author Response · Author response to Decision Letter 1]

12 Oct 2024

Dear Editor and Reviewers,

Firstly, thank you for the time and effort spent reviewing our manuscript. Your insightful comments and detailed feedback have been crucial in improving our work. We have carefully considered your suggestions and made the necessary revisions. We hope these changes meet your expectations and aid in the manuscript’s publication.

In this revision, we particularly emphasized the direct and indirect impacts of Environmental Information Disclosure (EID) on green technology innovation, explicitly revealing the mechanisms through which it operates alongside other environmental regulatory factors. Additionally, we have meticulously optimized the logical expression to ensure clarity and coherence in our discussion. Linguistically, we refined our wording based on the reviewers’ suggestions and conducted a thorough review and update of the reference formatting to meet academic standards.

To clearly demonstrate these modifications, we employed revision markings to visually indicate all the changes made. We hope these efforts meet the reviewers’ expectations and contribute to the acceptance and publication of our paper.

Next, we elaborate on the changes based on the reviewers’ suggestions

Reviewer #1: 

Author incorporate all the changes highlighted in comments. The author incorporate the changes and fix these in effective manner. Now the time to publish the author work I highly recommend it.

Response: Thank you for your positive feedback and for endorsing the publication of our work. We are grateful for your thorough review and guidance throughout the revision process. 

Reviewer #3: 

1.The innovation point of this article needs to be more precise. There is already some clear literature on the correlation between PITI and green technology innovation. The author needs to seek their unique features humbly.

Response: We revised the innovation points. The innovations were further summarized. Also only the value of the existing research was outlined. The specific modifications are as follows:

This study offers several key contributions. First, a multi-period DID model is employed to assess the overall impact of EID on enterprise green technology innovation. Previous studies have explored the relationship between EID and green innovation, often using single-period DID models [28]. This study, however, accounts for the effects of new market entrants and the relocation of existing firms, addressing a gap in prior research. Second, this study examines how EID influences the implementation of other environmental regulatory tools. While previous research has analyzed the impact of EID on technology innovation through mechanisms such as human capital, foreign direct investment [30], political pressures, enforcement channels [29], innovation environment, investment, talent [31], green innovation environment, industrial structure [16], and corporate social responsibility [28], the interaction between different regulatory tools remains underexplored. To fill this gap, the study investigates how EID enhances public participation-based, command-control, and market-incentive regulations, analyzing whether EID can strengthen other regulatory tools to promote green technology innovation.

2.For the multi period DID model, the construction of Model 1 in this article is incorrect and requires significant adjustments. As this article uses multi period DID, it’s crucial to provide rigorous evidence based on the current research progress of methods. The current recognition results are not reliable because only the case of classical DID is considered.

Response:We used a multi-temporal DID model, and we optimized the model and formulation based on the reviewers’ comments.

3.Why does local governments’ disclosure of environmental information directly lead to green technology innovation in enterprises? Whether in the theoretical hypothesis discussion in this article or in the empirical strategies later, this theory and logic have not provided satisfactory answers. According to relevant literature, the vast majority of enterprises are completely unaware of this PITI. If this article believes that PITI can affect the quality of green innovation, what is the specific path?

Response: We refined our original logic. Rewrote the hypothesis part of our research. And we have rewritten the logic of PITI affecting the green technology innovation of enterprises, summarizing in addition to the influential role of PITI. And, we rewrote the content of mechanism analysis. The specific modifications are as follows:

The existing research on the impact of environmental regulation on enterprise innovation forms the theoretical basis of this study. Neoclassical economic theory suggests that environmental regulations increase enterprises’ “compliance costs,” potentially hindering green innovation. In contrast, the “Porter Hypothesis” argues that well-designed regulations can encourage green technology innovation through an “innovation compensation” effect. Therefore, this study hypothesizes that EID positively influences enterprise green technology innovation.

Firstly, EID promotes enterprise green technology innovation through legitimacy pressures and reputation mechanisms. According to legitimacy theory, enterprises must follow social norms and environmental regulations to maintain their market legitimacy and reputation [42]. EID increases transparency by disclosing information about pollution and violations, subjecting enterprises to public and investor scrutiny. If the enterprise’s actions deviate from expectations, its legitimacy and market position may suffer, leading to potential losses in market share and investor trust [16]. As a result, enterprises need to adopt green technology that goes beyond regulatory requirements to build a responsible image [43].

Secondly, EID compels firms to engage in green technology innovation by internalizing externalities and maximizing long-term returns. Based on cost-benefit analysis theory, firms weigh investment costs against the direct and indirect benefits of innovation when making decisions regarding green technology investments. The transparency mechanism of EID directly transfers the costs of environmental pollution to enterprises, compelling them to assume greater environmental responsibilities [28]. In this context, enterprises consider the future costs of polluting behaviors and reassess the long-term benefits of green technology innovation. This reevaluation amplifies the perceived long-term gains from green technology innovation, encouraging firms to undertake green technology innovation activities.

Lastly, EID fosters industry-wide green technology advancement through competitive effects. By making environmental information transparent, EID shifts the focus of competition within industries from traditional factors like price and quality to environmental performance [44, 45]. Leading enterprises gain a “first-mover advantage” through green technology innovation, while other firms follow suit or pursue further innovation. This creates a demonstration and competitive effect within the industry, driving the overall upgrade and transition towards green technology.

4.In 3.3.3 Mechanism variable, both government-led environmental regulation and public participation-based environmental regulation are considered instrumental variables, which conflicts with the title.

Response: We are very sorry that we had a problem with the text, government-led environmental regulation and public participation-based environmental regulation are actually mechanism variables. And, we refine government-led environmental regulation into command-control and market-incentive environmental regulation. Thanks to the reviewer’s care, we have made the correction.

5.What is the basis for selecting public participation-based environmental regulation and government-led environmental regulation in the 4.4 Mechanism Analysis section, and which category should the PITI in this article belong to? We should not be enthusiastic about creating concepts but focus on rigorous argumentation.

Response: We are very sorry that our selection of mechanism variables appeared unsupported in the previous version. Now, we have supplemented it and added some literature to support the basis of the selection in this paper. We summarize the existing classification of environmental regulatory tools. According to the different implementation subjects of environmental regulatory tools, environmental regulatory tools can be categorized into two types: government-led environmental regulation, i.e., command-control environmental regulation, and market-incentive environmental regulation. The second is informal environmental regulation with non-government participation, including public participation-based environmental regulation, voluntary environmental regulation, and information disclosure mechanisms. This paper mainly explores the impact of environmental regulatory tools on green technology innovation from the outside of enterprises, so the voluntary environmental regulation of enterprises is outside the scope of this paper. In this paper, we focus on assessing the EID of cities through the Pollution Source Monitoring Information Disclosure Index (PITI). This type of environmental regulation belongs to the information disclosure mechanism.

The specific choices of command-control environmental regulation, market-incentive environmental regulation, and public participation-based environmental regulation are based on the following:

The first mechanism variable is public participation-based environmental regulation. Public participation-based environmental regulation emphasizes the voluntary actions of market participants in controlling pollution. Existing studies typically use the number of public complaints and environmental proposals related to environmental issues as measurement indicators [16,24,57]. Therefore, this study uses the natural logarithm of the number of environmental proposals in the city where the firm is located as a proxy for public participation-based environmental regulation.

The second mechanism variable is command-control environmental regulation. Command-control environmental regulation refers to government-imposed laws and environmental standards that limit corporate pollutant emissions to promote environmental protection. Prior literature commonly uses the number of environmental administrative penalties to measure the stringency of this regulation [48,57]. Similarly, this study uses the natural logarithm of the number of environmental administrative penalties (plus one) in the city where the firm is located to capture the intensity of command-control regulation.

The third mechanism variable is market-incentive environmental regulation. Market-incentive environmental regulation leverages market pricing mechanisms to incentivize firms to assume environmental responsibility and consider pollution control in their production processes. Scholars frequently use the amount of pollution fees as a key indicator of this regulation [24,58,59]. Following this approach, we employ the natural logarithm of the pollution fees in the city where the firm is located to measure the strength of market-incentive environmental regulation.

6.The citation of references in this article is inaccurate, and the language expression requires professional polishing to make the article more readable.

Response: We checked the formatting of the references through Google Scholar and made corrections for some grammar and wording.

Once again, thank you for your valuable feedback and guidance. We are grateful for the critiques and directions provided, which have significantly helped to deepen our research and refine our manuscript. We look forward to your assessment of these revisions and are more than willing to make further modifications to meet the journal’s standards for publication. Please do not hesitate to provide additional guidance.

Sincerely,

Xingqi Wang

School of Economics and Management

Shihezi University

xingqiwang1030@163.com

October 12, 2024

---

## [Decision Letter · Decision Letter 2]

16 Oct 2024

Does environmental information disclosure promote enterprise green technology innovation?

PONE-D-23-33221R2

Dear Dr. Wang,

We’re pleased to inform you that your manuscript has been judged scientifically suitable for publication and will be formally accepted for publication once it meets all outstanding technical requirements.

Kind regards,

Zhaoyang Zhao

Guest Editor

PLOS ONE

Additional Editor Comments (optional):

Reviewers' comments:

Reviewer's Responses to Questions

**Comments to the Author**

1. If the authors have adequately addressed your comments raised in a previous round of review and you feel that this manuscript is now acceptable for publication, you may indicate that here to bypass the “Comments to the Author” section, enter your conflict of interest statement in the “Confidential to Editor” section, and submit your "Accept" recommendation.

Reviewer #3: All comments have been addressed

2. Is the manuscript technically sound, and do the data support the conclusions?

Reviewer #3: Yes

3. Has the statistical analysis been performed appropriately and rigorously? 

Reviewer #3: Yes

4. Have the authors made all data underlying the findings in their manuscript fully available?

Reviewer #3: Yes

5. Is the manuscript presented in an intelligible fashion and written in standard English?

Reviewer #3: Yes

6. Review Comments to the Author

Reviewer #3: The author responded to the vast majority of the questions I raised in the previous version and currently has no major concerns.

7. PLOS authors have the option to publish the peer review history of their article (what does this mean?). If published, this will include your full peer review and any attached files.

Reviewer #3: No

---

## [Editor Report · Acceptance letter]

4 Dec 2024

PONE-D-23-33221R2 

PLOS ONE

Dear Dr. Wang, 

I'm pleased to inform you that your manuscript has been deemed suitable for publication in PLOS ONE. Congratulations! Your manuscript is now being handed over to our production team.

Kind regards, 

on behalf of

Dr. Zhaoyang Zhao 

Guest Editor

PLOS ONE